# Groundwater Hydrogeochemical and Quality Appraisal for Agriculture Irrigation in Greenbelt Area, Iraq

Eman Sh. Awad [1,2], Noor S. Imran [3], Muthanna M. Albayati [4], Vyacheslav Snegirev [1], Tamara M. Sabirova [1], Natalia A. Tretyakova [1], Qusay F. Alsalhy [5,*], Mustafa H. Al-Furaiji [6], Issam K. Salih [7] and Hasan Sh. Majdi [7]

1   Department of Chemical Technology of Fuel and Industrial Ecology, Institute of Chemical Technology, Ural Federal University Named after the First President of Russia B.N. Yeltsin, 620002 Yekaterinburg, Russia; ial-samarraii@urfu.ru (E.S.A.); v.a.snegirev@urfu.ru (V.S.); t.m.sabirova@urfu.ru (T.M.S.); n.a.tretyakova@urfu.ru (N.A.T.)
2   Environmental Research Center, University of Technology-Iraq, Baghdad 10066, Iraq
3   Department of Architecture Engineering, Al Safwa University College, Karbala 56001, Iraq; 10947@uotechnology.edu.iq
4   Department of Civil Engineering, University of Technology-Iraq, Baghdad 10066, Iraq; muthanna.m.abdulhameed@uotechnology.edu.iq
5   Membrane Technology Research Unit, Department of Chemical Engineering, University of Technology-Iraq, Baghdad 10066, Iraq
6   Environment and Water Directorate, Ministry of Science and Technology, Baghdad 10066, Iraq; m.h.o.alfuraiji@alumnus.utwente.nl
7   Department of Chemical Engineering and Petroleum Industries, Al-Mustaqbal University College, Babil 51001, Iraq; dr_issamkamil@mustaqbal-college.edu.iq (I.K.S.); dr.hasanshker@mustaqbal-college.edu.iq (H.S.M.)
*   Correspondence: qusay.f.abdulhameed@uotechnology.edu.iq

**Abstract:** This study highlights the groundwater hydrogeochemical characteristics and processes (hydrochemistry characteristics, ion exchange, and salinization) and quality suitability assessment for irrigation purposes from five wells in the Greenbelt area located in northwestern Al-Najaf Governorate, Iraq. The suitability of groundwater for irrigation was assessed based on the irrigation water quality index (IWQI) for thirteen parameters and groundwater quality indices such as total dissolved solids (TDS), electrical conductivity (EC), sodium adsorption ratio (SAR), soluble sodium percent (SSP), residual sodium carbonate (RSC), total hardness (TH), permeability index (PI), potential salinity (PS), Kelley's ratio (KR), and magnesium hazard ratio (MHR). The IWQI's average values ranged between 76–139. The results of IWQI for the first and second sampling sites showed values of 139 and 104, respectively, indicating that the groundwater was unsuitable and unsafe for irrigation. In contrast, the IWQI for the third, fourth, and fifth sites were 83, 97, and 76, respectively, indicating that the groundwater was safe and possibly used for irrigation. The EC, TDS, PS, and MHR indices were all found to be unsuitable for irrigation in all five sites, and the KR index was also found to be unsuitable for agricultural irrigation in about 80% of the sites, while it was found that the indices of SAR, SSP, RSC, PI, and TH for all sites were suitable and safe for irrigation. As a result of this study, it has been determined that groundwater in the study area is unsuitable for agricultural irrigation. For sustainable groundwater exploitation, it is advised that a continuous water-quality-monitoring program should be implemented, as well as the development of suitable management practices.

**Keywords:** groundwater; agriculture irrigation; SAR; IWQI; irrigation indices; ion exchange

## 1. Introduction

Groundwater is a natural resource and a significant water source in urban and sub-urban areas in Africa and Asia; it is used for domestic and agricultural purposes due to the low availability and quality of surface-water resources [1]. According to Al-Mussawi et al. [2], groundwater is an essential source of irrigation and human drinking water in Iraq, particularly in rural parts of the western desert.

Groundwater quality is affected by anthropogenic and natural activities. Natural factors such as geology and geochemical processes can impact groundwater quality. The increase in population, urbanization, and industrialization has significantly deteriorated groundwater quality due to its frequent contamination with sewage seepage, intrusion of industrial wastewater, and manure for weed growing [3–5].

Hydrogeochemical investigations that control the chemistry of groundwater aid in developing knowledge of hydrochemical systems. This, in turn, can help with the efficient use and sustainable management of groundwater resources by identifying relationships between various hydrogeological parameters [6]. Zhao et al. [7] collected and analyzed samples from shallow and deep aquifers to reveal hydrogeochemical characteristics of the groundwater using the Piper trilinear diagram.

The Water quality index (WQI) is a practical and comparably simple approach that uses a comprehensive set of water-quality data and converts it into a single value representing overall groundwater quality and suitability for drinking, irrigation, and other uses [8]. The score of the Irrigation Water Quality Index (IWQI) ranges between 0–100. When it is closer to zero, the water quality will be excellent and safe for using in irrigation; and vice versa—the closer to 100—the water quality will be unsuitable for agricultural and require proper treatment before usage [9]. There are few studies to assess the quality of groundwater in this region. Sataa et al. [10] studied the groundwater quality for five wells in the same area using the water quality index of the Canadian model (CCME-WQI) for irrigation purposes and a weighted arithmetic index method for drinking purposes. The authors found that almost all sites have high salinity, represented as EC, TDS, and $SO_4^{2-}$. The irrigation water quality index (IWQI) for wells were classified as poor and regarded as moderate-restriction, and can be used for irrigation, except for two wells, where it was found that they are severely restricted (IWQI > 100) and cannot be used due to the high oil content in them, resulting from the upstream of the Najaf refinery. While the water quality index (WQI) for drinking purposes showed three wells regarded as 'fair' and can be used in the case of the availability of treatment units, there are nevertheless two wells unfit for human use. In a similar study, Alikhan et al. [11] stated that the WQI of groundwater in the same area is classified under the 'poor' category, which influences human health and socioeconomic conditions. Using the water quality index, Pei-Yue et al. [12] determined the quality of groundwater in Pengyang, China using an information-entropy method for computing WQI 14 parameters. The excellent-quality groundwater area covered nearly 90% of the whole study area. As a result, groundwater can be used for irrigation, and with particular pretreatment can also be used for human drinking.

The chemical ions present in groundwater determine their suitability for different purposes. The assessment of water quality is of utmost significance, especially in areas that depend on groundwater [13]. Twigg [14] assessed the groundwater quality of five wells between Al-Kifel and Al-Najaf Governorates. The physical and chemical analysis for ten parameters found that the concentrations were minimal at some wells and increased in others due to the physical and chemical pollutants from agriculture and the service and industrial activities in this area, which are responsible for the well's pollution. Chung et al. [15] and Adebayo et al. [9] estimated the groundwater suitability for irrigation and domestic purposes using irrigation water quality indices such as SAR, Na %, PI, RSC, TH and MH. They found that the classification of groundwater suitability showed that the groundwater is safe and suitable for agricultural and domestic purposes.

The hydrogeochemical and quality assessment of groundwater based on the irrigation water quality index (IWQI), and other indices were conducted to evaluate the suitability of groundwater for agricultural irrigation purposes in the Greenbelt project (northwestern Al-Najaf Governorate, Iraq). The study objectives are (i) to understand the groundwater chemistry, (ii) to assess the quality of groundwater, and (iii) to determine the suitability of groundwater used for irrigation purposes.

## 2. Materials and Methods

### 2.1. Site Description (Location, Climate)

The Iraqi government established the Greenbelt project in 2006 to improve the environment in the area. It is one of the most important projects in Al-Najaf Governorate, located in southwestern Iraq about 160 km south of Baghdad (Capital of Iraq). The Greenbelt area is located in northwestern AL-Najaf Governorate on the highway between Al-Najaf and Karbala governorates. It has eight wells, and three of them are out of service.

Al-Najaf Governorate has a hot desert climate, with long and hot summers and cold winters. The weather conditions for the study area included mean air temperature of minimum 18.3 °C and maximum 32.1 °C, relative humidity of 41.8%, wind speed of 1.63 m/s, sunshine of 8.62 h/day, evaporation of 294.3 mm, and rainfall of 7.78 mm with an average for twenty years (1994–2014) as unpublished meteorological data, taken from the Iraqi Meteorological Organization and Seismology [10].

### 2.2. Geological and Hydrogeological Setting

The study area is part of the western plateau (the Najaf desert), bordered on the east by the Euphrates River and the alluvial plain, on the south by Lake Najaf, and on the north by the Karbala and Babil governorates. The study area consists of successive rock formations of alluvial origin, whose periods range between the Tertiary and the Quaternary periods. The rock formations of the study area include Dammam, Euphrates, Fatha, Injana, Zahra, and Dibdibba Formations, as follows [16]:

(1) Dammam Formation: This consists of limestone, chalk, and organic rocks. The Dammam Formation is deposited in a coastal and continental environment with warm, highly saline waters; (2) Euphrates Formation: This formation consists of chalky limestone and sandy limestones. Its thickness ranges from 10–16 m and the depositional environment is shallow and warm; (3) Fatha Formation: This consists of sandy and calcareous rocks, with a thickness between 10–15 m. The depositional environment of this formation is a coastal marine environment; (4) Injana Formation: This consists of a succession of clay rocks and layers of sandy rocks rich in calcareous carbonates, and its thickness is about 35 m. (5) Zahra Formation: This consists of a succession of limestone and clay rocks or sandy and clay rocks. Its thickness reaches about 30 m; (6) Dibdibba Formation: This consists of fragile sediments that include a mixture of sand and gravel derived from igneous rocks, and the thickness of the formation ranges from 2–10 m [16].

Two of the sampling sites are located within the Dammam Formation, while the other sites are located within the Dibdibba Formation.

The study area is considered one of the important areas from a hydrogeological view because it contains groundwater reservoirs represented by the sandy Dibdibba formations and limestone Dammam formation. The exposure of part of the sandy Dibdibba formation helps to renew its water and maintain groundwater storage in it through the penetration of rainwater and surface flood into it. The groundwater moves within the Dibdibba reservoir from the west towards the east and southeast in the region, and the quality of the groundwater in this reservoir is characterized by a high concentration of dissolved salts. The Dammam Formation is considered a water reservoir in most of the area, and the western hydraulic boundaries of this reservoir are areas of continuous groundwater movement coming from the west and southwest of the western desert. The groundwater in the Dammam limestone reservoir moves from the west towards Euphrates River. Its water is a mixture of old and newer water that comes from rainwater in the previous rainy periods; and the salinity of the groundwater in this reservoir is of variable concentration [17].

### 2.3. GeoDatabase

Samples of groundwater were collected from five selected wells (Table 1) in the Greenbelt area during 2015–2016. Figure 1 illustrates the location of groundwater-sampling sites. Table 1 presented the boundaries and information for each well (well depth (WD), static water level (SWL), dynamic water level (DWL), and discharge (D)).

**Table 1.** Coordinates of the groundwater-sampling sites by GPS (Garmin navigator).

| Well (Site No.) | X-Latitude | Y-Longitude | WD, m | SWL, m | DWL, m | Discharge, L/s |
|---|---|---|---|---|---|---|
| S1 | 32 9 59.64° | 44 11 46.02° | 183 | 48 | 45 | 7 |
| S2 | 32 9 56.7° | 44 14 7.08° | 172 | 25 | 48 | 9 |
| S3 | 32 9 58.2° | 44 15 41.28° | 50 | 24.9 | 32.1 | 6 |
| S4 | 32 10 22.02° | 44 16 46.32° | 50 | 16 | 22 | 7 |
| S5 | 32 10 46.26° | 44 17 58.68° | 50 | 18 | 25 | 9 |

Well depth (WD), static water level (SWL), dynamic water level (DWL), and discharge (D).

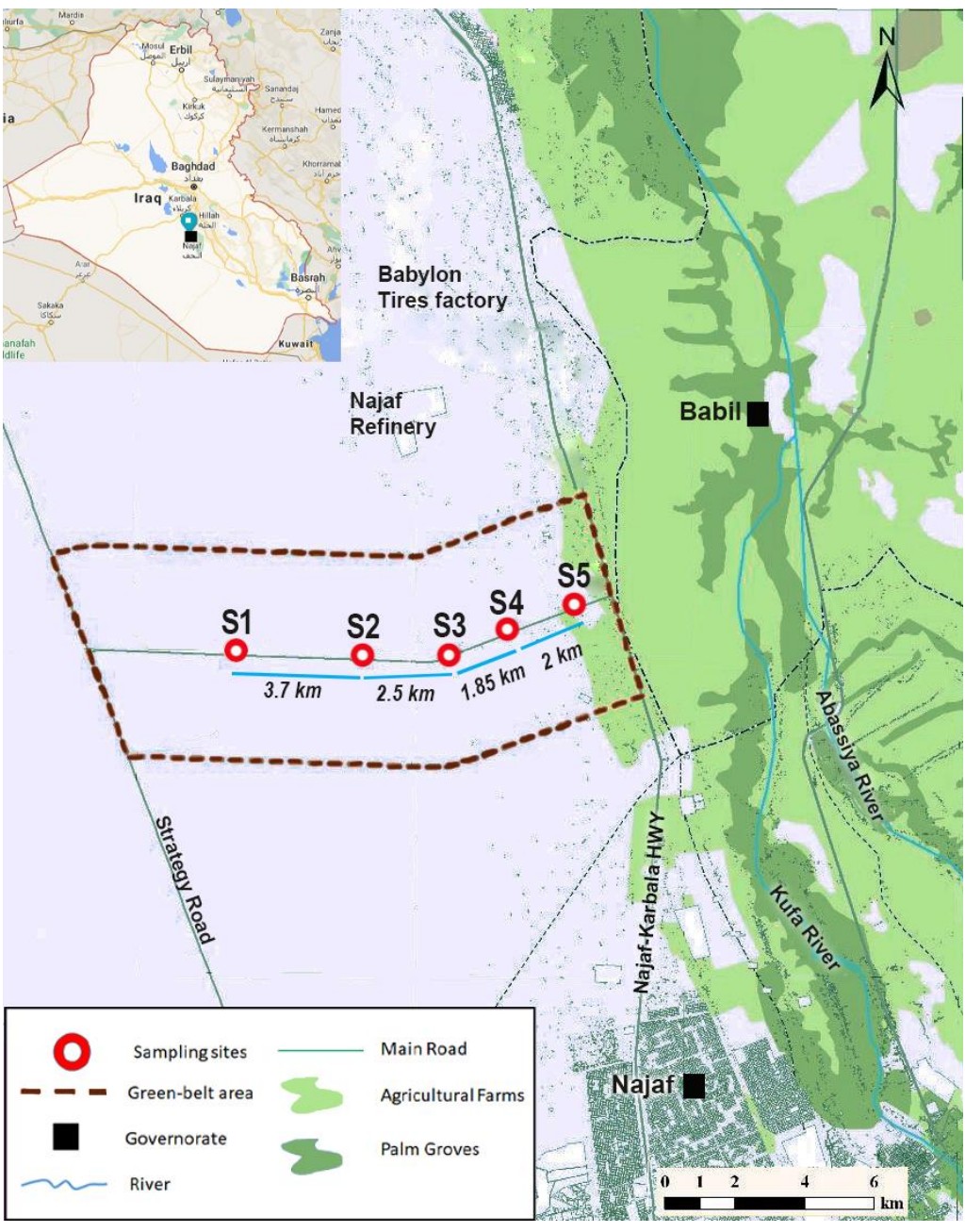

**Figure 1.** The groundwater-sampling sites at the Greenbelt area, northwestern Al-Najaf Governorate, Iraq.

Groundwater samples were preserved in stopper-fitted polyethylene bottles and were kept in an icebox during transportation to the laboratory. Conductivity and pH were measured on the sites by using a portable device pH/EC/TDS meter (HANNA HI9811-5).

All parameters were analyzed according to the standard methods for analyzing water and wastewater [18]. Total dissolved solids (TDS), carbonate ($CO_3^{2-}$), bicarbonate ($HCO_3^-$), chloride ($Cl^-$), magnesium ($Mg^{2+}$), and calcium ($Ca^{2+}$) were analyzed by titration methods; potassium ($K^+$) and sodium ($Na^+$) were tested using flame photometric method by industrial flame photometer (PFP7); while phosphate ($PO_4^{3-}$), nitrate ($NO_3^+$), and sulfate ($SO_4^{2-}$) were analyzed by using a multiparameter photometer (HANNA HI 83200).

*2.4. Assessment Methods*

2.4.1. Ion Exchange

Base Exchange Index is proposed by Schoeller [19], which is measured by using Chloro-alkaline indices CAI I and CAI II through rock–water interaction. Chloro-alkaline indices are used to investigate the ion exchange between the groundwater and the aquifer, which occurred in the chemical reaction during the movement and rest state of water. It is calculated by the formulae

$$CAI\ I = \left(Cl^- - \left(Na^+ + K^+\right)\right)/Cl^-, \tag{1}$$

$$CAI\ II = \left(Cl^- - \left(Na^+ + K^+\right)\right)/\left(SO_4^{2-} + CO_3^{2-} + HCO_3^- + NO_3^-\right) \tag{2}$$

where all ions are expressed in meq/L.

2.4.2. Salinization

Revelle index (RI) was used to assess the salinization level of groundwater in the study area. The Revelle Index was calculated using the formula below [20]:

$$RI = Cl^-/(HCO_3^- + CO_3^{2-}) \tag{3}$$

The ions' concentrations are stated in meq/L in the equation above.

2.4.3. Irrigation Water Quality Index (IWQI)

The water quality index method presents a cumulatively numerical expression specifying a given level of water quality that is extensively utilized worldwide because of its ability to represent water-quality information properly. It is one of the most valuable indicators and crucial parameters for assessing and managing groundwater quality [12,21]. The weighted arithmetic module was used to calculate the irrigation water quality index (IWQI) for thirteen parameters in this study. This approach gives data on water-quality evaluations and is calculated using the following formula [22]:

$$IWQI = \frac{\sum W_i Q_i}{\sum W_i} \tag{4}$$

where $Q_i$ is the quality-relative value of $i^{th}$ parameters, which represents the number of water-quality parameters; $W_i$ is a unit weight which measures the importance of a parameter in the calculation of the *WQI* index; $Q_i$ and $W_i$ are calculated by the following formulas:

$$Q_i = \frac{V_i - V_o}{S_i - V_o} * 100 \tag{5}$$

$$W_i = \frac{K}{S_i} \tag{6}$$

where $V_i$ represents the value experimentally obtained from laboratory analysis; $V_o$ represents the ideal value of the water-quality parameter, the ideal value of all parameters taken

as zero, excepting pH value counted as 7; $S_i$ represents the standard guidelines of water quality for agriculture [23]; K is a constant which can be obtained from $K = 1/\sum(1/S_i)$. The classification of the water quality status based on IWQI [9,24] is illustrated in Table 2.

**Table 2.** Classification of water quality according IWQI score.

| Score | <25 | 26–50 | 51–75 | 76–100 | >100 |
|---|---|---|---|---|---|
| IWQI class | Excellent | Good | Poor | Very poor | Unsuitable |
| Possible usage | Drinking, irrigation, and industrial | Drinking, irrigation, and industrial | irrigation and industrial | irrigation | Proper treatment required before use. |
| | SAFE | | | | UNSAFE |

### 2.4.4. Suitability for Irrigation

Different irrigation indices are determined to find the suitability. The irrigation indices include total dissolved solids (TDS), electrical conductivity (EC), sodium adsorption ratio (SAR), soluble sodium percent (SSP), residual sodium carbonate (RSC), total hardness (TH), permeability index (PI) potential salinity (PS), Kelley's ratio (KR), and magnesium hazard ratio (MHR). Irrigation water quality indices for the groundwater used in this study are summarized in Table 3. In calculating these indices, all the concentrations of ions were expressed in milli-equivalent per liter (meq/L).

**Table 3.** The formula of irrigation water quality indices.

| Indices | Formula | Ref. |
|---|---|---|
| SAR | $SAR = \dfrac{Na^+}{\sqrt{\frac{Ca^{2+}+Mg^{2+}}{2}}}$ | [25] |
| SSP | $SSP = \dfrac{Na^+}{Ca^{2+}+Mg^{2+}+Na^++K^+} * 100$ | [26] |
| RSC | $RSC = \left(CO_3^{2-} + HCO_3^-\right) - \left(Ca^{2+} + Mg^{2+}\right)$ | [27] |
| PS | $PS = Cl^- + \left(0.5 * SO_4^{2-}\right)$ | [28] |
| KR | $KR = \dfrac{Na^+}{Ca^{2+}+Mg^{2+}}$ | [29] |
| MH | $MH = \dfrac{Mg^{2+}}{Ca^{2+}+Mg^{2+}} * 100$ | [30] |
| TH | $TH = 2.50 * Ca^{2+} + 4.12 * Mg^{2+}$ | [30] |
| PI | $PI = \dfrac{Na^+ + \sqrt{HCO_3^-}}{(Na^+ + Ca^{2+} + Mg^{2+})} * 100$ | [28] |

## 3. Results and Discussion

### 3.1. Hydrogeochemical Processes

3.1.1. Hydrochemistry Characteristics

Descriptive statistical analyses of groundwater parameters obtained from field measurements and laboratory tests were calculated, including the mean, minimum, maximum, and standard deviation by using of SPSS 28.0 software. The statistical analyses are tabulated in Table 4. Physicochemical parameters are pH, electrical conductivity (EC), total dissolved solids (TDS), and the individual cations and anions.

The pH of the groundwater samples is slightly basic (6.9–8.0), with an average of 7.34, indicating good water quality in terms of pH [31].

The analyzed samples have high values of EC with a mean value of 5026 μS/cm, exceeding the permissible limit values (EC 3000 μS/cm) of the water quality for agriculture guidelines [23]. Moreover, the high EC values indicate that some form of mixing between freshwater and saline water occurs, as proposed by Egbi et al. [32]. The TDS has a mean value of 3216 mg/L, exceeding the guideline value of TDS, 2000 mg/L [23], indicating that the obtained groundwater samples are highly salinized.

**Table 4.** Statistical summary of the physicochemical parameters of groundwater samples.

| Parameters | Units | Minimum | Maximum | Mean | Std. Deviation |
|---|---|---|---|---|---|
| pH | - | 6.9 | 8.0 | $7.34 \pm 0.2$ | 0.46 |
| EC | μs/cm | 4680 | 5960 | $5026 \pm 238$ | 534.12 |
| TDS | mg/L | 2995 | 3814 | $3216.56 \pm 153$ | 341.87 |
| $Ca^{2+}$ | mg/L | 180.68 | 471.34 | $333.077 \pm 48$ | 108.40 |
| $Mg^{2+}$ | mg/L | 453.78 | 1089.00 | $669.659 \pm 113$ | 252.30 |
| $Na^+$ | mg/L | 400 | 490 | $447.6 \pm 18$ | 39.89 |
| $K^+$ | mg/L | 32.8 | 46.0 | $38.56 \pm 2.33$ | 5.21 |
| $HCO_3^-$ | mg/L | 12.20 | 31.73 | $22.926 \pm 4.13$ | 9.23 |
| $Cl^-$ | mg/L | 969.5 | 3278.0 | $1861.3 \pm 527$ | 1179.00 |
| $PO_4^{3-}$ | mg/L | 0.01 | 0.12 | $0.068 \pm 0.02$ | 0.04 |
| $NO_3^-$ | mg/L | 0.000 | 63.435 | $40.9504 \pm 12.44$ | 27.81 |
| $SO_4^{2-}$ | mg/L | 1040 | 1940 | $1512 \pm 148$ | 330.33 |

The order of abundance of the major cations (in mg/L) was according to the following decreasing order: $Na^+ > Ca^{2+} > Mg^{2+} > K^+$ for the groundwater of sampling site S1; $Mg^{2+} > Na^+ > Ca^{2+} > K^+$ for sampling site S2; and $Mg^{2+} > Na^+ > Ca^{2+} > K^+$ for sampling sites S3, S4, and S5. The abundance of the major cations for all sites is presented in Figure 2. Magnesium ($Mg^{2+}$) is the dominating cation in most of the sample sites, varying between 453.78 to 1089 mg/L (mean value 669.65 mg/L). Along with $Mg^{2+}$, both sodium ($Na^+$) (400–490 mg/L) and calcium ($Ca^{2+}$) (180.68–471.34 mg/L) contributed in considerable amounts to the mineralogical composition of the samples. Potassium ($K^+$) shows a range of 32.8–46 mg/L with an average of 38.65 mg/L.

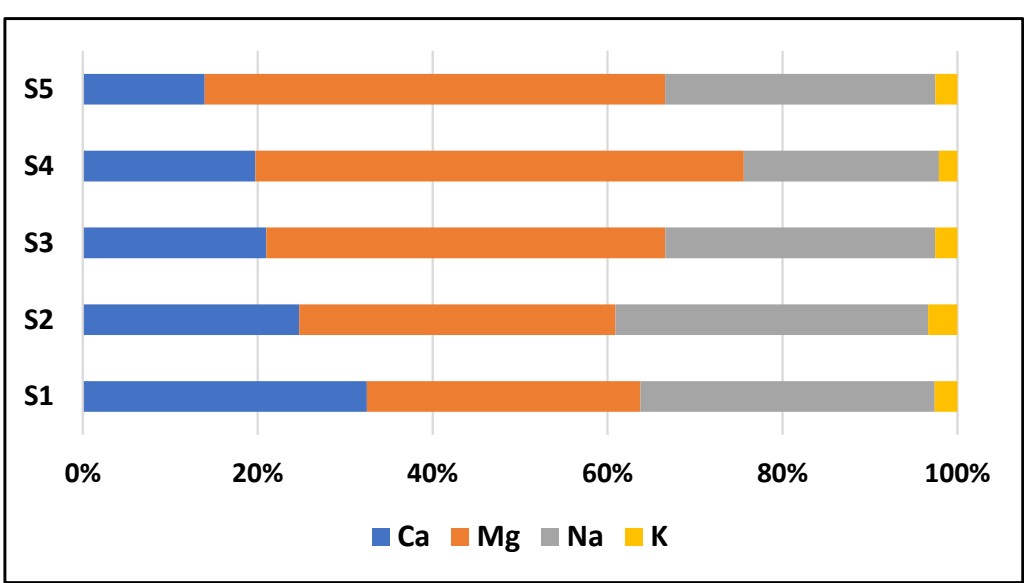

**Figure 2.** The abundance of the major cations for all sites.

The order of concentration of the major anions (in mg/L) showed the following decreasing order: $SO_4^{2-} > Cl^- > HCO_3^- > NO_3^- > PO_4^{3-}$ for the samples of both S1 and S4; $SO_4^{2-} > Cl^- > NO_3^- > HCO_3^- > PO_4^{3-}$ for the samples of both S2 and S3; while for sampling site S5 it was $Cl^- > SO_4^{2-} > NO_3^- > HCO_3^- > PO_4^{3-}$. The abundance of the major anions for all sites is presented in Figure 3. Chloride ($Cl^-$) is the dominant anion for all sampling sites except S1 with a range of 969.5–3278 mg/L, while sulfate ($SO_4^{2-}$) is the dominating anion only for sampling site S1, varying in the range of 1040–1940 mg/L, while the other anions were in low concentrations.

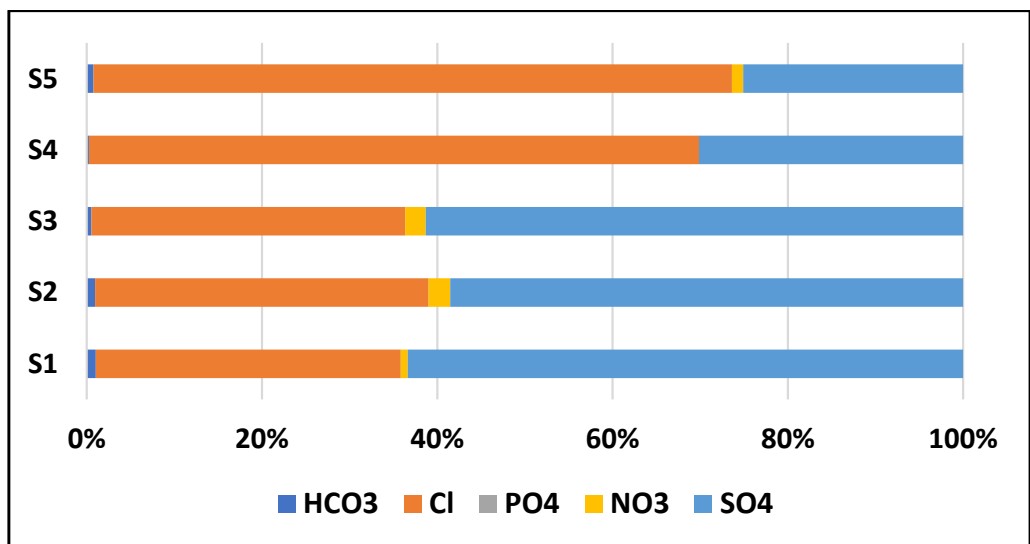

**Figure 3.** The abundance of the major anions for all sites.

In terms of the major cations and anions, most samples' measured parameters satisfy the standard water-quality guidelines for agriculture [23] except $Mg^{2+}$ and $K^+$, which is more than the allowable limit for agriculture irrigation for all sites. At the same time, the $Cl^-$ is more than the allowable limit only in sampling sites S4 and S5.

The reason for the increase in the concentration of ions is due to several factors: The direct effect of rocks and soil on the concentration of groundwater ions, and through the geology of the study area, it was noted that it consists of gypsum and anhydrite rocks. Therefore, the inflow rainwater may dissolve the salts and turn them to the groundwater [10].

Infiltrating water also affects the quality and quantity of groundwater. Rainwater Infiltration may increase groundwater contamination, particularly in areas with sandy soils and shallow streams, pollutants may not have the opportunity to decompose or sorb onto soil particles, so they will transfer with the inflowing water into the groundwater [33].

In addition, there are no restrictions on the use of fertilizers, inefficient irrigation methods, as well as human activities. All of these reasons increase the concentration of ions and affect the quality of groundwater [11].

### 3.1.2. Ion Exchange

The indices CAI I and II will be positive or negative based on the exchange process between ($K^+$ and $Na^+$) from the aquifer material with ($Ca^{2+}$ and $Mg^{2+}$) in groundwater and vice versa. It would be negative when calcium or magnesium in groundwater is exchanged with sodium or potassium in the aquifer material. However, when sodium or potassium in groundwater is exchanged with calcium or magnesium in aquifer material, the above indices would be positive, indicating reverse ion exchange [19,34]. The indices CAI I and II estimated for samples showed a positive ratio, indicating the dominance in this study's groundwater of ion exchange.

The relationship between ($Ca^{2+} + Mg^{2+}$) and ($SO_4^{2-} + HCO3^-$) in reverse ion exchange will be near 1:1 aquiline, indicating the minerals' dissolution, as shown in Figure 4. Furthermore, the relationship between ($Na^+ - Cl^-$) and ($Ca^{2+} + Mg^{2+} - HCO_3^- - SO_4^{2+}$) can be used to identify the reverse ion exchange. According to Fisher et al., 1997 [35], the relationship will be linear, with a ($-1$) slope. In this study, the groundwater samples were plotted in a linear fashion, as shown in Figure 5, and the slope was $-0.146$. As a result, it is clear that reverse ion exchange is one of the key mechanisms governing a chemistry of the region's groundwater.

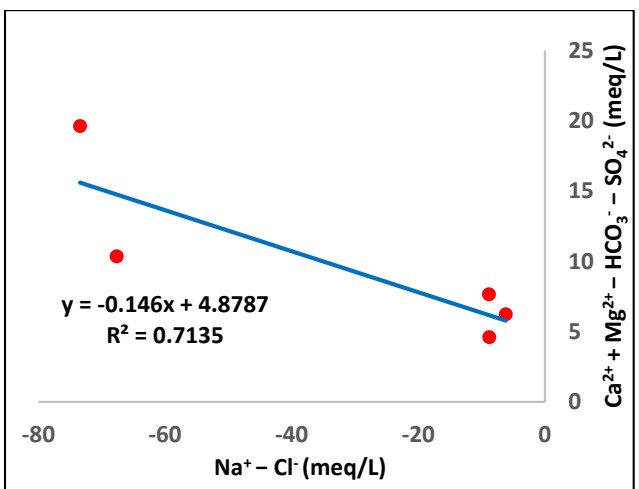 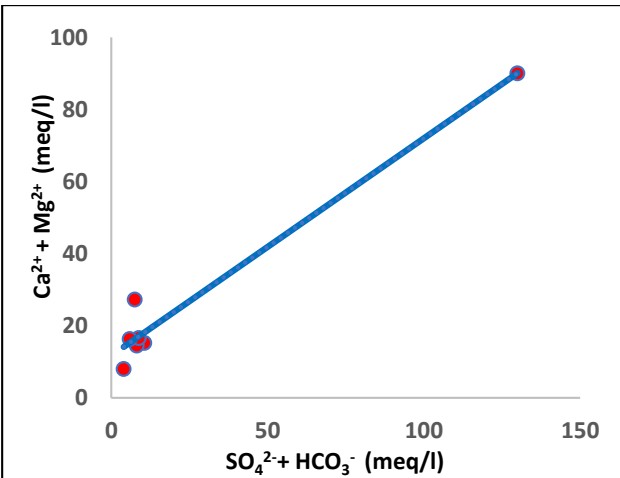

**Figure 4.** Plot of numerous ions, representing reverse ion exchange of groundwater samples.

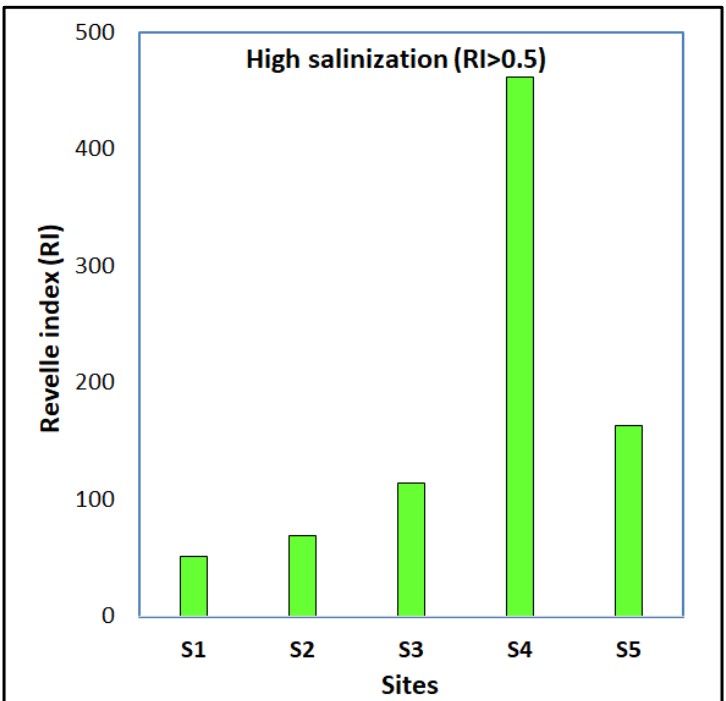

**Figure 5.** The Revelle index (RI) for the groundwater samples.

### 3.1.3. Salinization

The groundwater is affected by salinization with RI values > 0.5 [34]. The estimated RI values ranged from 51.75 to 461.69 (Figure 5). All the samples of groundwater have very high RI values, which indicates that they are affected by high salinization. This is attributed to geogenic activities in this study area, caused by anthropogenic inputs such as domestic waste, unrestricted activities of agriculture such as pollution with agricultural wastewater, and septic-tank seepage, which lead to a significant increase in chloride-ion concentration in these groundwater wells [31].

### 3.2. Groundwater-Quality Assessment

In this study, the IWQI was applied for all sites using the standard water-quality guidelines for irrigation [23]. The first step in the 'weighted arithmetic index' method for the calculation of IWQI includes the determination of 'unit weight' assigned for each physic-

ochemical parameter used in the calculation. The highest unit weight, 0.84, is assigned for carbonate ($CO_3^{2-}$) and 0.042 for both potassium ($K^+$) and phosphate-P ($PO_4$-P), indicating the importance of these indicators in water-quality assessment and their significant impact on the index. Table 5 shows the standards (permissible values of several parameters) for the groundwater as well as the unit weights assigned to each parameter considered in the calculation of IWQI. The detected values of selected physicochemical parameters in all sites of groundwater and the values of WQI are shown in Table 6.

**Table 5.** The standard water-quality guidelines for irrigation and unit weights for parameters.

| Parameters | Units | Agriculture Standard [23] | Unit Weights ($W_i$) |
|---|---|---|---|
| pH | - | 7.25 | 0.011594 |
| Total dissolved solid (TDS) | mg/L | 2000 | 0.000042 |
| Carbonate ($CO_3^{2-}$) | meq/L | 0.1 | 0.840600 |
| Bicarbonate ($HCO_3^-$) | meq/L | 10 | 0.008406 |
| Chloride ($Cl^-$) | meq/L | 30 | 0.002802 |
| Calcium ($Ca^{2+}$) | meq/L | 20 | 0.004203 |
| Magnesium ($Mg^{2+}$) | meq/L | 5 | 0.016812 |
| Sodium ($Na^+$) | meq/L | 40 | 0.002102 |
| Potassium ($K^+$) | mg/L | 2 | 0.042030 |
| Phosphate-P ($PO_4$-P) | mg/L | 2 | 0.042030 |
| Nitrate-N ($NO_3^+$-N) | mg/L | 10 | 0.008406 |
| Ammonium-N ($NH_4^+$-N) | mg/L | 5 | 0.016812 |
| Sulfate ($SO_4^{2-}$) | meq/L | 20 | 0.004203 |
| $\sum W_i$ | | | 1.0000 |

**Table 6.** The values of the physicochemical parameters and the WQI values in all the groundwater sites.

| Scheme 32. | Para. | pH | TDS | $CO_3^{2-}$ | $HCO_3^-$ | $Cl^-$ | $Ca^{2+}$ | $Mg^{2+}$ | $Na^+$ | $K^+$ | $PO_4$-P | $NO_3^+$-N | $NH_4^+$-N | $SO_4^{2-}$ | WQI |
|---|---|---|---|---|---|---|---|---|---|---|---|---|---|---|---|
| S1 | $V_i$ | 8.00 | 3186.20 | 0.06 | 0.52 | 30.00 | 5.89 | 9.34 | 21.22 | 37.80 | 0.03 | 5.60 | 1.16 | 12.13 | 139 |
| | $Q_i$ | 400 | 159.31 | 60.00 | 5.20 | 100.00 | 29.46 | 186.7 | 53.04 | 1890.0 | 1.30 | 56.00 | 23.30 | 60.63 | |
| | $Q_iW_i$ | 4.64 | 0.01 | 50.44 | 0.04 | 0.28 | 0.12 | 3.14 | 0.11 | 79.44 | 0.05 | 0.47 | 0.39 | 0.25 | |
| S2 | $V_i$ | 7.20 | 2994.20 | 0.00 | 0.40 | 27.48 | 4.24 | 10.20 | 21.30 | 46.00 | 0.02 | 14.55 | 2.07 | 9.38 | 103 |
| | $Q_i$ | 80.0 | 149.71 | 0.00 | 4.00 | 91.60 | 21.21 | 203.9 | 53.26 | 2300.0 | 1.14 | 145.45 | 41.47 | 46.88 | |
| | $Q_iW_i$ | 0.93 | 0.01 | 0.00 | 0.03 | 0.26 | 0.09 | 3.43 | 0.11 | 96.67 | 0.05 | 1.22 | 0.70 | 0.20 | |
| S3 | $V_i$ | 7.60 | 3039.00 | 0.00 | 0.24 | 27.31 | 3.61 | 12.93 | 18.48 | 35.00 | 0.04 | 14.30 | 1.03 | 10.38 | 82 |
| | $Q_i$ | 240 | 151.95 | 0.00 | 2.39 | 91.03 | 18.07 | 258.5 | 46.20 | 1750.0 | 1.96 | 143.00 | 20.50 | 51.88 | |
| | $Q_iW_i$ | 2.78 | 0.01 | 0.00 | 0.02 | 0.26 | 0.08 | 4.35 | 0.10 | 73.55 | 0.08 | 1.20 | 0.34 | 0.22 | |
| S4 | $V_i$ | 7.00 | 3813.40 | 0.00 | 0.20 | 92.34 | 4.81 | 22.41 | 18.91 | 41.20 | 0.02 | 0.00 | 5.41 | 8.88 | 97 |
| | $Q_i$ | 0.00 | 190.67 | 0.00 | 2.00 | 307.79 | 24.06 | 448.2 | 47.28 | 2060.0 | 0.98 | 0.00 | 108.25 | 44.38 | |
| | $Q_iW_i$ | 0.00 | 0.01 | 0.00 | 0.02 | 0.86 | 0.10 | 7.53 | 0.10 | 86.58 | 0.04 | 0.00 | 1.82 | 0.19 | |
| S5 | $V_i$ | 7.00 | 3045.40 | 0.00 | 0.52 | 85.03 | 2.26 | 14.03 | 17.39 | 32.80 | 0.00 | 12.00 | 1.22 | 6.50 | 76 |
| | $Q_i$ | 0.00 | 152.27 | 0.00 | 5.20 | 283.43 | 11.29 | 280.5 | 43.48 | 1640.0 | 0.16 | 120.00 | 24.38 | 32.50 | |
| | $Q_iW_i$ | 0.00 | 0.01 | 0.00 | 0.04 | 0.79 | 0.05 | 4.72 | 0.09 | 68.93 | 0.01 | 1.01 | 0.41 | 0.14 | |

Average values of IWQIs ranged from 76 to 139. The results of IWQI showed values of 139 and 104 in sampling sites S1 and S2, respectively, indicating that the water quality index for irrigation was considered an 'unsuitable' category (>100) and unsafe for irrigation. This may be due to the presence of the Najaf Refinery, which is about 7 km from the study area; as there is no treatment for discharge resulting from the refining operations, it is the main source of pollution. Wastewater is also disposed directly to the land; thus, the waste is discharged through the soil layers and reaches the groundwater. Therefore, public awareness must be raised not to use water of this quality; the highest priority should be given to regular water-quality monitoring. Appropriate technologies should be employed to make this groundwater more suitable for irrigation. The Al-Najaf Refinery affects the wells in sampling sites S1 and S2 more than the others. The values for sampling sites S3 and S4 were 83 and 97; they are observed to be in the 'very poor' category (76–100), while site S5 showed a value of 76, which indicates a 'poor' category (51–75). All of the last three sites can be used for irrigation with high-permeability soil. Fortunately, the study area has

high-permeability sandy soil, which is located in the desert of Al-Najaf. The summary of IWQI values of the groundwater samples from the five sampling sites is shown in Figure 6.

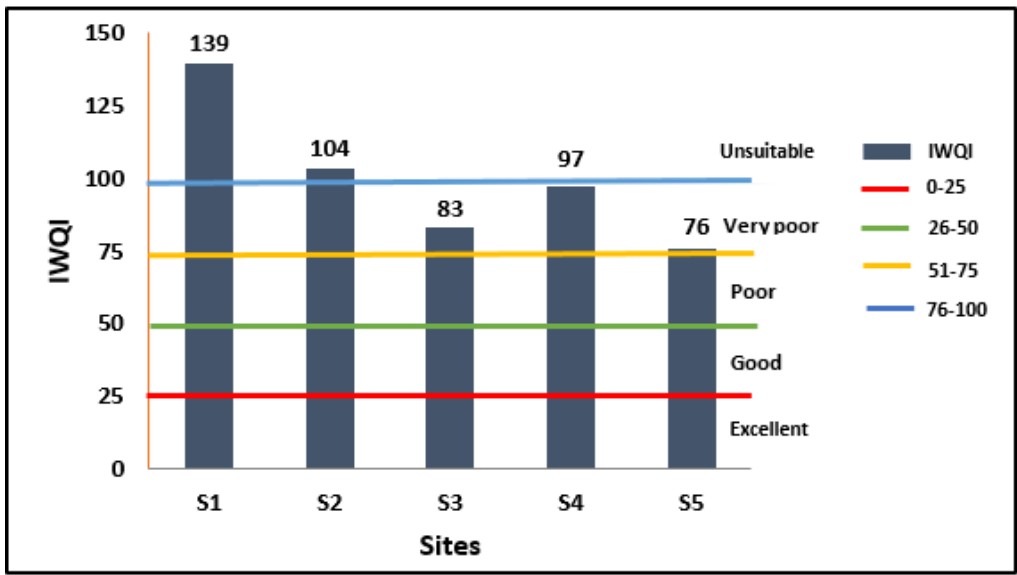

**Figure 6.** The values of IWQI of the groundwater samples.

*3.3. Groundwater Suitability for Irrigation*

In the study region, the groundwater suitability for agricultural irrigation is presented by comparing the analytical results of indices with corresponding parameters of irrigation-water classifications (Table 7) [36–38].

**Table 7.** Suitability classification for agricultural irrigation from groundwater.

| Indices | Range | Classification | Distribution% |
|---|---|---|---|
| Electrical Conductivity (EC) | <250 | Excellent | - |
| | 250–750 | Good | - |
| | 750–2000 | Doubtful | - |
| | >2000 | Unsuitable | 100% |
| Sodium Adsorption Ratio (SAR) | <10 | Excellent | 100% |
| | 10–18 | Good | - |
| | 18–26 | Doubtful | - |
| | >26 | Unsuitable | - |
| Soluble Sodium Percent (SSP) | <20 | Excellent | - |
| | 20–40 | Good | - |
| | 40–60 | Permissible | 80% |
| | 60–80 | Doubtful | 20% |
| | >80 | Unsuitable | - |
| Residual Sodium Carbonate (RSC) | <1.25 | Good | 100% |
| | 1.25–2.5 | Doubtful | - |
| | >2.5 | Unsuitable | - |
| Total Hardness (TH) | <60 | Soft | 40% |
| | 61–120 | Moderately hard | 60% |
| | 121–180 | Hard | - |
| | >180 | Very hard | - |
| Permeability Index (PI) | Class I (>75) | Excellent | - |
| | Class II (25–75) | Good | 100% |
| | Class III (<25) | Poor | - |
| Potential Salinity (PS) | <3 | Suitable | - |
| | >3 | Unsuitable | 100% |
| Kelley's Ratio (KR) | <1 | Suitable | 20% |
| | >1 | Unsuitable | 80% |
| Magnesium Hazard Ratio (MHR) | <50 | Suitable | - |
| | >50 | Unsuitable | 100% |

### 3.3.1. Salinity Hazard

Salinity is an essential parameter in establishing water suitability for irrigation usage. It is usually measured as the electrical conductivity (EC) or the total dissolved solids (TDS) in the water. They are both essential interlinked parameters for water quality in irrigation [36]. EC is a measure of the degree to which water conducts electricity; it measures the salinity hazard to crops [38]. The EC values of groundwater samples varied from 4680 to 5960 µS/cm (mean value 5026 µS/cm).

In contrast, TDS values ranged from 2995 to 3814 mg/L (mean value 3216 mg/L), as shown in Figure 7, indicating a severe degree of restriction on using this groundwater in irrigation (Table 8) [23]. It is obvious that irrigation with saline water will increase the concentration of salt in soil, and this might be a problem if the concentration rises to a level that is harmful to the crops, because plants depend directly on the salinity of water and every crop has a limit of salinity; therefore, the groundwater for these sites may be used with crops that have tolerance to high levels of salt such as sunflower, rye, wheat, and olive. As a result, while using groundwater for irrigation, it is necessary to keep the salinity under control.

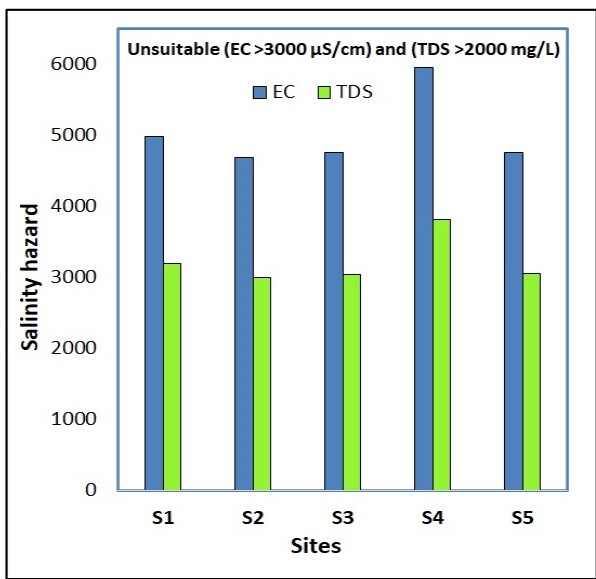

**Figure 7.** The salinity hazard (EC and TDS) values for the groundwater samples.

**Table 8.** Restriction degree for agriculture irrigation.

| Parameters | None | Slight to Moderate | Sever |
|---|---|---|---|
| EC, μS/cm | <700 | 700–3000 | >3000 |
| TDS, mg/L | <450 | 450–2000 | >2000 |

### 3.3.2. Sodium Hazard

Sodium hazard is commonly stated as a sodium adsorption ratio (SAR), which is computed by comparing the proportion of sodium ($Na^+$) ions to the concentration of calcium ($Ca^{2+}$) and magnesium ($Mg^{2+}$) [7]. SAR is one of the valuable parameters for evaluating the suitability of groundwater in irrigation because it is responsible for the sodium hazard. Because SAR was concerned with salt absorption on the soil surface, it provides a good foundation for evaluating the sodium damage degree in irrigation water. Moreover, sodium from irrigation water reaches the soil and replaces the calcium and magnesium that were absorbed, resulting in a reduction in permeability and poor drainage in the soil [39,40]. The SAR values of the samples vary from 5.13 to 7.93 meq/L (Figure 8A). All the groundwater samples belonged to the excellent category according to the SAR classification of irrigation water (Table 7), indicating that the groundwater can be suitable for all soil types.

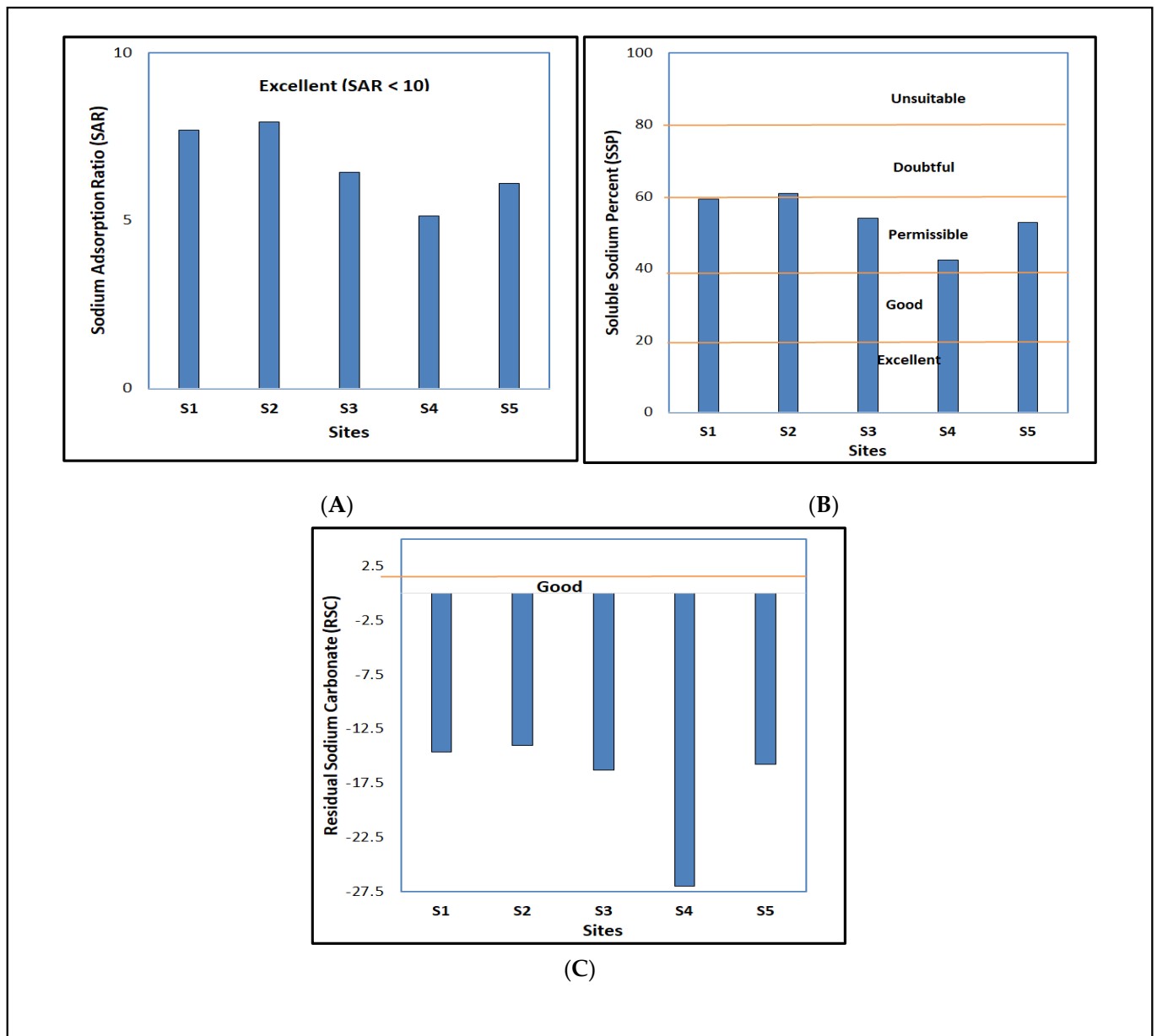

**Figure 8.** The sodium hazard for the groundwater samples. (**A**) Sodium adsorption ratio (SAR) < 10, excellent; (**B**) Soluble sodium percentage (SSP) < 60, permissible; (**C**) Residual sodium carbonate (RSC) < 1.25, Good.

In order to better correctly and intuitively measure irrigation-water quality in this research, EC was considered together with SAR, as shown in Table 9, which is based on the combined effect of EC (salinity hazard) and SAR (sodium hazard) [41]. All collected groundwater samples are in class C4-S1, indicating highly saline water, and they are not appropriate for irrigation under normal conditions. Although this sort of water can be used for plants with good salt tolerance, it restricts the suitability for irrigation, especially in soils with limited drainage [42].

**Table 9.** Classification of irrigation water for salinity hazard and sodium hazard.

| Salinity Hazard Class | EC, μS/cm | TDS, mg/L | Classes | Characteristics |
|---|---|---|---|---|
| C1 | 0–250 | <200 | Excellent | Low-salinity water can be utilized to irrigate most soils with little risk of soil salinity developing. |
| C2 | 251–750 | 200–500 | Good | Medium-salinity water can be used for irrigation if there is a moderate quantity of drainage. |
| C3 | 751–2250 | 501–1500 | Permissible | High-salinity water should not be used on soil with poor drainage. Even with adequate drainage, salinity control may necessitate particular management. |
| C4 | >2250 | 1501–3000 | Unsuitable | Very high-salinity water is not suitable for irrigation under normal conditions. |
| **Sodium hazard class** | **SAR meq/L** | **Irrigation-water suitability** | | **Characteristics** |
| S1 | 0–10 | Low | | Suitable for all types of soils except for crops that are particularly sensitive to sodium. |
| S2 | 10–18 | Medium | | Suitable for organic or coarse-textured soils with good permeability. In fine-textured soil, it is unsuitable. |
| S3 | 18–26 | High | | Harmful for almost all types of soils. Good drainage, high leaching, and gypsum addition are all required. |
| S4 | >26 | Very high | | Unsuitable for irrigation. |

The soluble sodium percentage (SSP) is an important parameter to assess water suitability for irrigation purposes. According to the US Department of Agriculture [43], the standard SSP value is between 40–60%, and the SSP classification of irrigation water is presented in Table 7. The SSP values of the collected samples range from 42.32% to 60.89% (Figure 8B). All the groundwater samples are found in the good-to-permissible zone. However, groundwater samples from station S2 tend to shift towards the permissible-to-doubtful zone for irrigation purposes. The groundwater in the study area is suitable for irrigation, since it does not impair soil permeability due to reactions with the soil.

Residual sodium carbonate (RSC) was computed to estimate the hazardous influence of carbonate ($CO_3^{2-}$) and bicarbonate ($HCO_3^-$) on water quality for agricultural irrigation usage. The excess concentration of $CO_3^{2-}$ and $HCO_3^-$ ions in groundwater at higher levels than the concentrations of $Ca^{2+} + Mg^{2+}$ ions causes precipitation of $Ca^{2+}$ and $Mg^{2+}$ and influences unsuitability for irrigation [44]. Water with RSC < 1.25 meq/L (low) is safe and suitable for irrigation purposes, whereas it is marginally suitable up to 2.5 meq/L (medium) and unsuitable for irrigation over 2.5 meq/L (high), as recommended by the US Salinity Laboratory [42]. The RSC classification of irrigation water is presented in Table 7. The RSC values varied from −27.02 to −14.04 (Figure 8C) with an average value of −17.55, indicating that all the collected samples were categorized as safe for irrigation purposes in terms of RSC.

### 3.3.3. Total Hardness (TH)

The water hardness results from the presence of divalent cations such as $Ca^{2+}$ and $Mg^{2+}$; the dissolution of calcium and magnesium is the primary cause of total hardness. A moderate TH value is beneficial for protecting the plumbing system from corrosion. The acceptable limit of TH is 100 mg/L, which provides adequate corrosion control. TH is usually classified as follows: (0–60 mg/L) as soft; (60–120 mg/L) as moderately hard; (120–180 mg/L) as hard; and (>180 mg/L) as very hard [45,46]. Total hardness varied between 52.62 and 104.35 mg/L (Figure 9A), and the TH classification of irrigation water is presented in Table 7. The groundwater samples from stations S1 and S2 are grouped in

the soft water category. At the same time, the rest of them fall under the moderately hard water category, showing suitability for water-supply systems (pipes) in irrigation.

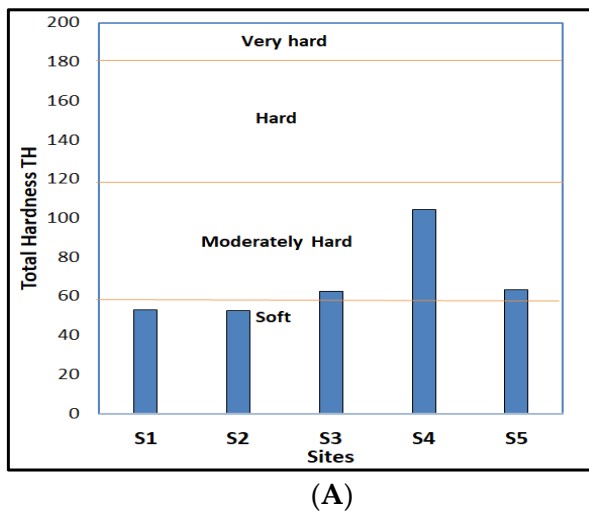
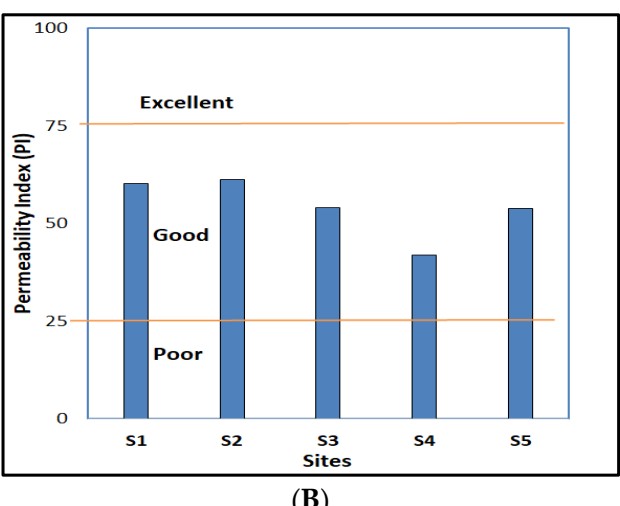

**Figure 9.** Total hardness and permeability for the groundwater samples. (**A**). Total hardness (TH) < 60, soft; 61–120, moderately hard; (**B**). Permeability (PI) 25–75, good.

3.3.4. Permeability Index (PI)

According to a significant impact of the levels of Na, Mg, Ca, and bicarbonate in irrigation water on soil permeability during long-term use, the permeability index (PI) was utilized in this study to assess groundwater suitability for irrigation. From the analysis for the samples and calculations, it was found that the values of PI are 60.16, 61.37, 54.17, 41.97, and 53.78 meq/L (Figure 9B). According to the Doneen classification [28] (Table 7), all samples fall under Class II (25 < PI < 75), which considered that groundwater samples from these sites were not of an excellent rating but good (moderately suitable) for long-term agricultural irrigation with little effect on the property of soil, and recommended some cautions when it comes to soil permeability [9,47].

3.3.5. Potential Salinity (PS)

Potential salinity (PS) is another parameter index for water quality. It is a chloride- and sulfate-dominant index for categorizing water for irrigation purposes. A PS value less than 3 meq/L shows that the water can be used for irrigation [28,48]. The PS values of all collected samples were estimated at high values (Table 7); the maximum PS value was 96.04 meq/L, and the average PS value was 56.37 meq/L (Figure 10A). This is due to the high levels of $SO_4^{2+}$ and $Cl^-$ in the groundwater of the study area. In terms of PS, all of the samples were determined to be unsuitable for agricultural irrigation (PS > 3 meq/L).

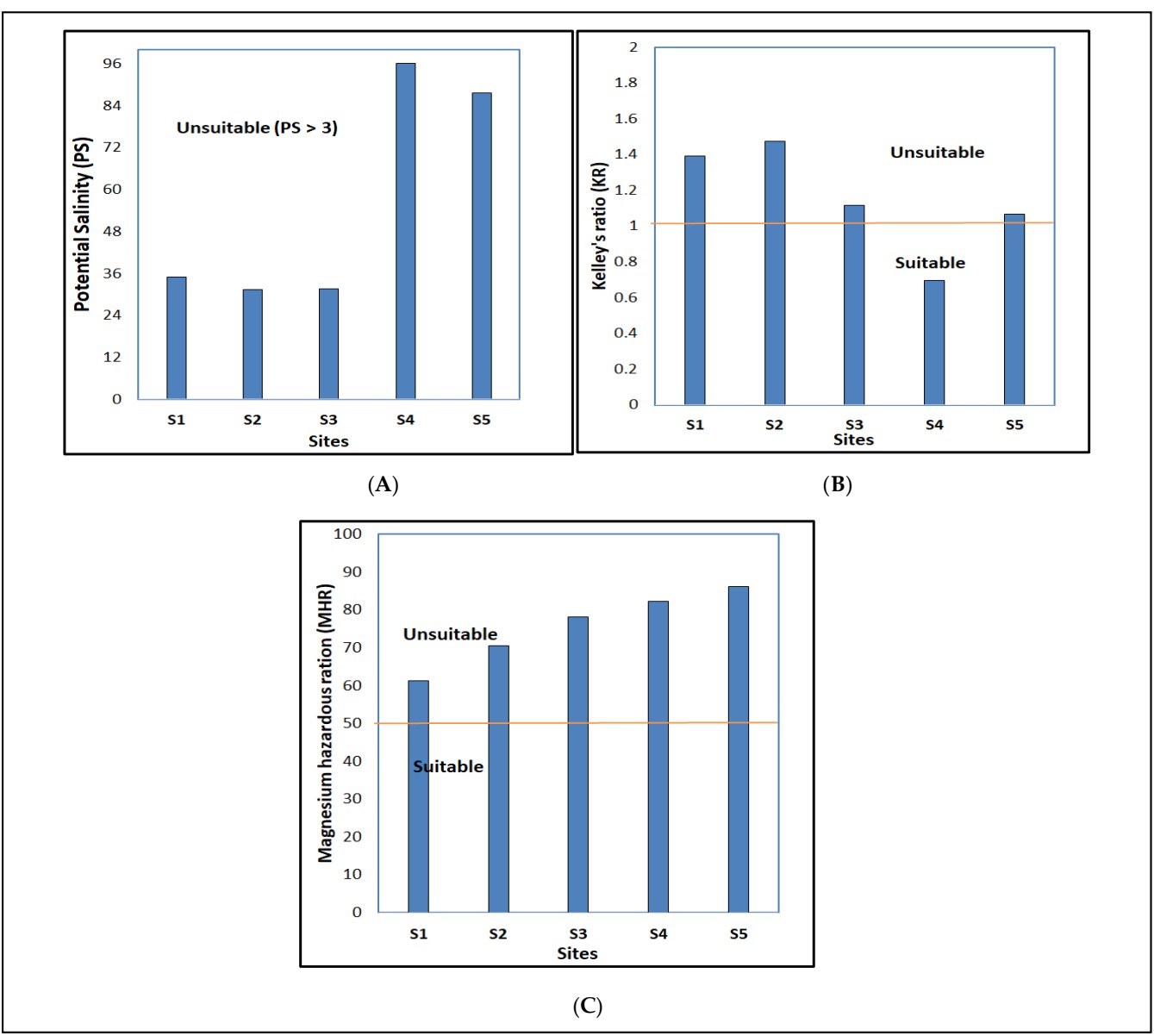

**Figure 10.** Irrigation quality indices for the groundwater samples. (**A**). Potential salinity (PS) > 3, unsuitable; (**B**). Kelley's ratio (KR) < 1 suitable, KR > 1 unsuitable; (**C**). Magnesium hazard ratio (MHR) < 50 suitable, MHR > 50 unsuitable.

3.3.6. Kelley's Ratio (KR)

Kelley's Ratio is an essential index used to assess the suitability of groundwater for irrigation purposes in terms of main cations (i.e., $Na^+$, $Mg^{2+}$, and $Ca^{2+}$). The Kelley Index value of 1 indicates that sodium concentration should be almost equal to the sum of calcium and magnesium, resulting in a perfect balance. Waters with a KR value < 1 indicate a sodium deficit and are suitable for irrigation, whereas those with a higher ratio indicate excess sodium and are unsuitable [29].

KR showed values varying between 1.48 and 0.69 with an average value of 1.15 (Figure 10B). All the groundwater samples showed KR > 1 except one sample, S4, which showed KR < 1. Water from sampling station S4 has relatively low sodium-ion content than other samples, which can be used for irrigation purposes, while the other samples are unsuitable for irrigation. The KR classification of irrigation water is presented in Table 7.

### 3.3.7. Magnesium Hazard

One of the most significant parameters for irrigation water-quality assessment is the magnesium hazard ratio (MHR). $Ca^{2+}$ and $Mg^{2+}$ are generally in equilibrium in most waterways. Groundwater with a high $Mg^{2+}$ concentration would interchange with $Na^+$ in the soil., causing soil alkalization and reducing crop yield [7,49]. All of the groundwater samples showed MHR values of 61.31, 70.62, 78.15, 82.32, and 86.13 for the groundwater samples from stations S1 to S5, respectively (Figure 10C). These values were greater than 50 meq/L, indicating that all samples were unsuitable for irrigation purposes (Table 7), This could lead to soil magnesium alkalization as a result of long-term groundwater irrigation.

## 4. Conclusions

A comprehensive analysis of the chemical composition and hydrogeochemical features was conducted to determine the suitability of groundwater for irrigation purposes, by using the irrigation water quality index (IWQI), multiple indices, and a multiaspect assessment. The sequence of the abundance of the major cations was in the following order: $Na^+ > Ca^{2+} > Mg^{2+} > K^+$ for groundwater of sampling site S1; $Mg^{2+} > Na^+ > Ca^{2+} > K^+$ for sampling site S2; and $Mg^{2+} > Na^+ > Ca^{2+} > K^+$ for sampling sites S3, S4, and S5. Meanwhile, for the major anions the sequence was in the following order: $SO_4^{2-} > Cl^- > HCO_3^- > NO_3^- > PO_4^{3-}$ for the samples of both S1 and S4; $SO_4^{2-} > Cl^- > NO_3^- > HCO_3^- > PO_4^{3-}$ for the samples of both S2 and S3; while for the sampling site S5 as $Cl^- > SO_4^{2-} > NO_3^- > HCO_3^- > PO_4^{3-}$. The chemical composition of groundwater was mostly controlled by ion exchange, which are the dominant processes in the study area.

According to the IWQI values, the groundwater samples fall from 'poor' to the 'unsuitable' category in sites S1 and S2 of the study area, indicating unsafe irrigation, while sites S3 and S4 fall within the 'very poor' category (76–100), and S5 falls within the 'poor' category (51–75). All the last three sites are safe and possibly used for irrigation. The groundwater suitability for irrigation was assessed based on the irrigation quality indices including EC, SAR, SSP, RSC, TH, PI, PS, KR, and MHR. The extremely high EC, TDS, and PS values indicated that all of the samples were inappropriate. unsuitable category. About 80% of samples with KR content fell in the unsuitable category, except groundwater from sampling site S4, which was suitable for irrigation. The high MHR % values indicated that all groundwater samples were unsuitable for irrigation. However, the samples evaluated depending on the index of SAR, SSP, RSC, and PI were suitable for irrigation. Moreover, the TH of samples fell within the 'soft' to 'moderately soft' category for all groundwater sites. Using groundwater in the study area as a source of irrigation would impose significant salinity hazards, while the degree of sodium hazards was moderately low. All of the samples fall within the C4S1 sector according to the salinity and sodium hazard classification. This is due to the long period of residence of water, minerals' dissolution from the rock structure, and the intrusion of domestic and industrial wastewater.

The findings of this study will help planners and policymakers establish a strategy to handle comparable issues in other places, as well as being beneficial for sustainable management of groundwater resources in the studied area.

**Author Contributions:** Conceptualization, E.S.A., N.S.I., Q.F.A. and V.S.; Methodology, E.S.A., N.S.I., V.S., Q.F.A. and N.A.T.; Validation, Q.F.A., T.M.S., M.H.A.-F. and H.S.M.; Formal analysis, E.S.A., N.S.I., V.S. and Q.F.A.; Investigation, T.M.S., N.A.T., Q.F.A., I.K.S. and H.S.M.; Data curation, E.S.A., Q.F.A., M.M.A. and M.H.A.-F.; Writing—original draft, E.S.A. and Q.F.A.; Writing—review & editing, E.S.A., Q.F.A. and M.H.A.-F.; Visualization, Q.F.A.; Supervision, T.M.S., N.A.T. and Q.F.A.; Funding acquisition, E.S.A. and Q.F.A. All authors have read and agreed to the published version of the manuscript.

**Funding:** This research received no external funding.

**Institutional Review Board Statement:** Not applicable.

**Informed Consent Statement:** Not applicable.

**Data Availability Statement:** Not applicable.

**Conflicts of Interest:** The authors declare no conflict of interest.

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
