# Peer review of "Groundwater Hydrogeochemical and Quality Appraisal for Agriculture Irrigation in Greenbelt Area, Iraq"

_environments, doi:10.3390/environments9040043_

Round 1

Reviewer 1 Report

This MS deal with evaluation on the groundwater possible use for irrigation based on indices IWQI, TDS, EC, SAR,... Results show some key indices, including IWQI and TDS, are higher than the standard of water quality, and therefore, not suitable for irrigation. Though the results are clear based on the water quality of groundwater, there are some shortcomings in this MS that should be considered in the revising process:

  • Based on the standard of irrigation water quality, this ground water is unsuitable for irrigation. However,, for some salt tolerated crops, this water can be partially used. Then, the plants is the main factor, that determine the irrigation water quality. In some paper, water of TDS of 2-5g/l can also be used for irrigation. This should be discussed.
  • There is no description on the five well position. In the figure. The five wells are close. Therefore, we can think they are one site, and cannot represent the whole region water quality status.
  • There is no explanations on the change of water quality among the five sites. And also the reason for this water quality. This should be discussed.
  • he conclusion and result in this section need more discussion. (1) ions distribution in groundwater is mainly influenced by evaporation when water table is high, which is generally 2-3 m below ground surface. Now the water table is 16-38 m in table 1, therefore, the effect of evaporation on groundwater quality could be much low. (2) groundwater quality also is greatly influenced by inflow water and infiltration water quality. this should be discussed. (3) ions in soil are also contributed more to the groundwater quality. I suggest to use these materials in discussion section.
  • There are too many equations in results section, that should be moved to material section.
  • Subsection “3.1.2 evaporation” is not directly related to the main topic of this study, this part should be regard as discussion and put in the appropriate position.
  • The position of the greenbelt project in the study region should be positioned on the figure. Also the position of refined factory and all possible factors that influence the groundwater quality should be showed on the figure. the elevation map should be merged into this map, which can be used to show the surface water flow in the study region. Further, the activities of greenbelt project in this region, soil physical and chemical properties should be provided, which also important to analyze the ground water quality dynamics in the study region. the spacing between well should be described.

Other comments can be found in the pasted file with tracked comments.

Author Response

Cover Letter

Dear Editor,

I would like to submit a revised form of the manuscript entitled "Groundwater Hydrogeochemical and Quality Appraisal for Agriculture Irrigation in Greenbelt Area, Iraq" for consideration for publication in the Environments. This article is revised according to the reviewers’ comments and highlighted in different color in the revised manuscript. Please answer to the reviewers’ comments as in the attached report.

I appreciate the effort of the Reviewers to improve our article, thank you.

Your consideration for this manuscript with revised form is highly appreciated.

Sincerely

Prof. Dr. Qusay F. Alsalhy

Membrane Technology Research Unit

Chemical Engineering Department

University of Technology,

Alsinaa Street No. 52

Baghdad, Iraq

      [email protected]

Comments and Suggestions for Authors

Reviewer 1

This MS deal with evaluation on the groundwater possible use for irrigation based on indices IWQI, TDS, EC, SAR,... Results show some key indices, including IWQI and TDS, are higher than the standard of water quality, and therefore, not suitable for irrigation. Though the results are clear based on the water quality of groundwater, there are some shortcomings in this MS that should be considered in the revising process:

  • Based on the standard of irrigation water quality, this ground water is unsuitable for irrigation. However, for some salt tolerated crops, this water can be partially used. Then, the plants is the main factor, that determine the irrigation water quality. In some paper, water of TDS of 2-5g/l can also be used for irrigation. This should be discussed.

Answer: Thank you for your comment:

It’s obvious that irrigation with saline water will increase the concentration of salt in soil, and this might be a problem if the concentration rises to a level that is harmful to the crops, because of plants depended directly on the salinity of water and every crop has a limit of salinity, therefore the groundwater for these sites may be used with the crops that tolerance to high level of salt like Sunflower, Rye, Wheat, and Olive. As a result, while using groundwater for irrigation, it is needful to keep the salinity under control.

  • There is no description on the five well position. In the figure. The five wells are close. Therefore, we can think they are one site, and cannot represent the whole region water quality status.

Answer: Done. Thank you

  • There is no explanations on the change of water quality among the five sites. And also the reason for this water quality. This should be discussed.

Answer: Thank you for your suggestion

Average values of IWQIs ranged from 76 to 139. The results of IWQI showed values of 139 and 104 in sampling sites S1 and S2, respectively, indicating that the water quality index for irrigation was considered an unsuitable category (>100) and unsafe for irrigation. This may be due to the presence of the Najaf Refinery, which is about 7 kilometers from the study area, as there is no treatment for discharge resulting from the refining operations, so it's the main source of pollution. Wastewater is also disposed directly to the land; thus, the waste is discharged through the soil layers and reaches the groundwater. Therefore, public awareness must be raised not to use water of this quality; the highest priority should be given to regular water quality monitoring. Appropriate technologies should be employed to make this groundwater more suitable for irrigation.  Thus, public awareness must be raised not to use water of this quality; the highest priority should be given to regular water quality monitoring. Appropriate technologies should be employed to make this groundwater more suitable for irrigation. The Al-Najaf Refinery effects on the wells in sampling sites S1 and S2 more than the others. The values for sampling sites S3 and S4were 83 and 97; they are observed in the very poor category (76-100), while site S5 showed a value of 76, which indicates a poor category (51-75). All the last three sites can be used for irrigation with the high permeability soil. Fortunately, the study area has the high permeability sandy soil which located in the desert of AL-Najaf.

  • The conclusion and result in this section need more discussion. (1) ions distribution in groundwater is mainly influenced by evaporation when water table is high, which is generally 2-3 m below ground surface. Now the water table is 16-38 m in table 1, therefore, the effect of evaporation on groundwater quality could be much low. (2) groundwater quality also is greatly influenced by inflow water and infiltration water quality. this should be discussed. (3) ions in soil are also contributed more to the groundwater quality. I suggest to use these materials in discussion section.

Answer:

We added in section 3.1.1.:

The reason for the increase in the concentration of ions is due to several factors: The direct effect of rocks and soil on the concentration of groundwater ions, and through the geology of the study area, it was noted that it consists of gypsum and anhydrite rocks. Therefore, the inflow rainwater may dissolve the salts and turn them to the groundwater [10].

Infiltrating water also affects the quality and quantity of groundwater. Rainwater Infiltration may increase groundwater contamination, particularly in areas with sandy soils and shallow streams, pollutants may not have the opportunity to decompose or sorb onto soil particles, so they will transfer with the inflowing water into the groundwater [32].

 In addition, there are no restrictions on the use of fertilizers, inefficient irrigation method as well as human activities. All of these reasons increase the concentration of ions and affect the quality of groundwater [11].

  • There are too many equations in results section, that should be moved to material section.

Answer:

Equations in results section moved to material section.

  • Subsection “3.1.2 evaporation” is not directly related to the main topic of this study, this part should be regard as discussion and put in the appropriate position.

Answer:

Depend on your suggestion that "the subsection “3.1.2 evaporation” isn't directly related to the main topic of this study", and because of "the effect of evaporation on groundwater quality could be much low in study area", therefore we decided to delete this section.

  • The position of the greenbelt project in the study region should be positioned on the figure. Also the position of refined factory and all possible factors that influence the groundwater quality should be showed on the figure. the elevation map should be merged into this map, which can be used to show the surface water flow in the study region. Further, the activities of greenbelt project in this region, soil physical and chemical properties should be provided, which also important to analyze the ground water quality dynamics in the study region. the spacing between well should be described.

Done, Thank you

Other comments can be found in the pasted file with tracked comments.

Page 1, line 24 & 25:

The IWQIs average values ranged between 76-139. The results of IWQI for 1st and 2nd sampling sites showed values of 139 and 104, respectively, indicating the groundwater were unsuitable and unsafe for irrigation. In contrast, the IWQI for the others sites 3rd, 4th and 5th were 83, 97, and 76, respectively, indicating the groundwater were safe and possibly used for irrigation.

Page 2 line 51:

We revised and added:

The score of Irrigation Water Quality Index (IWQI) ranges between 0-100. when it's closer to zero, the water quality will be Excellent and safe for using in irrigation , and vice versa, the closer to 100, the water quality will be unsuitable for agricultural and require a proper treatment before usage [9].

Page 2 line 91:

We added:

The weather conditions for the study area were include mean air temperature of minimum 18.3oC and maximum 32.1oC, relative humidity of 41.8%, wind speed of 1.63 m/s, the sunshine of 8.62 hr/day, evaporation of 294.3 mm, and rain (7.78 mm) with an average for twenty years (1994-2014) as unpublished meteorological data, took from the Iraqi meteorological organization and seismology [16] [10].

Page 3 line 106:

Done, Thank you

Page 3 line 108:

Done, Thank you

Page 4 line 118:

This is sentences in abstract:

The suitability of groundwater for irrigation was assessed based on the irrigation 20 water quality index (IWQI) for thirteen parameters.

And we added in line 116:

The weighted arithmetic module was used to calculate the irrigation water quality index (IWQI) for thirteen parameters in this study.

Page 4 line 123:

The agriculture standard guideline was listed in Table 5.

Page 4 line 127:

Done, Thank you

Page 5 line 175:

Units was added, and the mean value for each parameter among the five sites listed in Table 4.

Page 7 line 190:

The part moved to Material section.

Page 7 line 213:

Thank you for remark. There is a mistake in the figure. We revised it

Page 8 line 214:

We checked the data of this Figure, it’s right.

Page 8 line 215:

We revise it as follows:

All the samples of groundwater have very high RI values, which indicates that they are affected by high salinization. This attributed to geogenic activities in this study area, caused by anthropogenic inputs like domestic waste, unrestricted activities of agriculture like pollution with agricultural wastewater, and septic tank seepage that led to increase the chloride ion concentration significantly in these groundwater wells

Page 8 line 216:

The part moved to Material section.

Page 8 line 220:

The units were deleted (this is mistake), thank you.

About the values are true because the chloride values in these wells are very high due to waste, animal waste, and pollution with agricultural wastewater that led to increase in chloride ion concentration in groundwater.

Reviewer 2

  • The research topic is relevant for the readership of Environment. However, the paper needs more elaboration before it can be finally published. It lacks a clear strategy for motivation of the work and the methodology is not entirely clear. Some figures need additional revision.

Answer: Thank you for your suggestion please see the following:

We added more details to the paper, edited the methodology and revision figures.

The weather conditions for the study area were include mean air temperature of minimum 18.3oC and maximum 32.1oC, relative humidity of 41.8%, wind speed of 1.63 m/s, the sunshine of 8.62 hr/day, evaporation of 294.3 mm, and rain (7.78 mm) with an average for twenty years (1994-2014) as unpublished meteorological data, took from the Iraqi meteorological organization and seismology [16] [10].

The study area is part of the western plateau (the Najaf desert), bordered on the east by the Euphrates River and the alluvial plain, on the south by Lake Najaf, and on the north by the Karbala and Babil governorates. The study area consists of successive rock formations of alluvial origin, whose periods range between the tertiary and the Quaternary periods. The rock formations include: Dammam, Euphrates, Fatha, Injana, Zahra, and Dibdibba Formation. Two of the sampling sites are located within the Dammam Formation, which consists of limestone, chalk and organic rocks, while the rest of the sites are located within the Dibdibba Formation, which is composed of brittle materials, clays and sandstones [17].

2.3. Assessment methods

2.3.1. Ion-Exchange

Base Exchange Index proposed by Schoeller [19], which is measured by using Chloro-alkaline indices CAI I and CAI II through rock–water interaction. Chloro-alkaline indices are used to investigate the ion-exchange between the groundwater and the aquifer, which has occurred in the chemical reaction during the movement and rest state of water. It is calculated by the formulae:

 ,

(1)

(2)

Where all ions are expressed in meq/L.

2.3.2. Salinization

Revelle index (RI) was used to assess the salinization level of groundwater in the study area. The Revelle Index was calculated using the formula below [20]:

RI = Cl- / (HCO3- +CO32-)

(3)

The ions' concentrations are stated in meq/L in the equation above.

  • The presentation of the work needs improvement. The background information provided in the introduction section is too general. It is a general discussion of groundwater quality assessment, with references to many papers, but it is not focused on the specific topics analysed in the paper or the methods that are applied. In its current drafting, it does not help the reader to understand the motivation of the work and identify where the authors go beyond the state of the art with their work. I suggest rewriting this section, stating the main research questions addressed in the work and referencing the authors that have tackled these questions before, summarizing their results and identifying the gaps to be filled by this work.

Answer:

We edited and added:

  • Sataa et al [10] studied the groundwater quality for five wells in the same area using the water quality index of the Canadian model (CCME-WQI) for irrigation purpose and a weighted arithmetic index method for drinking purpose. The authors found almost all sites have high salinity, representing as EC, TDS and SO42-. The irrigation water quality index (IWQI) for wells were classified as poor and regarded as moderate restriction and can using for irrigation, except for two wells, it was found that they are severely restricted (IWQI>100) and can't be used due to the high oil content in it resulting from the upstream of the Najaf refinery. While water quality index (WQI) for drinking purpose showed three wells regarded as fair category and can be used in the case of the availability of treatment units, nevertheless, two wells unfit for human uses.
  • Using the water quality index, Pei-Yue et al. [12] determined the quality of groundwater in Pengyang, China using an information entropy method for computing WQI 14 parameters. The excellent-quality groundwater area covered nearly 90% of the whole study area. As a result, groundwater can be used for irrigation and also with particular pretreatment, can be used for human drinking.
  • Twigg [14] assessed the groundwater quality of five wells between Al-Kifel and Al-Najaf Governorates. The physical and chemical analysis for ten parameters found that the concentrations minimal at some wells and increased in others due to the physicals and Chemicals pollutants from agriculture, the service and industrial activities in this area which are the responsible for wells pollution.

  • I found the presentation of methodology a bit confusing. The methodology section presents the methods used, but sections 2.3.1 and 2.3.2 have the same subtitle. The authors should clearly state their objectives at the beginning and devote the methodological section to present their strategy, indicating how they are going to tackle the research questions addressed in their work and why they chose the methods they are proposing.

Answer:

We revise the methodology part and edited. Also, we transfer another section for it:

(2.3.1. Ion-Exchange) and (2.3.2. Salinization).

The other section changed to (Irrigation water quality index (IWQI)) and (Suitability for irrigation).

  • In addition, the manuscript is not properly finished. There are too many grammar and typographical errors that indicate that the manuscript is still at an early stage of revision. For instance, there are space (line 21) and extra punctuation (line 22) in the middle of the sentence. The authors should have checked their manuscript for such evident errors before submission. You may find below several of such errors, but the list is far from complete.

The manuscript should be polished.

Answer:

The manuscript has been revised and corrected.

Specific comments

Page 1 line 26 to 28 There is ambiguity in the expression of the sentence.

The EC, TDS, PS, and MHR indices were all found to be unsuitable for irrigation in all five sites, and the KR index was also found to be unsuitable for agricultural irrigation in about 80% of the sites. While it was found that the indices of SAR, SSP, RSC, PI, and TH for all sites were suitable and safe for irrigation.

Page 1 line 32 Keywords are words or terms selected from reports and papers for literature indexing to express the full text topic content information, rather than the selection of individual target indicators.

Done. Thank you

Page 2 line 63 The meaning of the text ‘The high-quality water covered nearly 90% of the region’ is not clear. Do you mean the study area?

We edited:

The excellent-quality groundwater area covered nearly 90% of the whole study area.

It is better to add hydrogeological conditions of the study area in Section 2.1 Site Description.

We added to section 2.1:

The study area is part of the western plateau (the Najaf desert), bordered on the east by the Euphrates River and the alluvial plain, on the south by Lake Najaf, and on the north by the Karbala and Babil governorates. The study area consists of successive rock formations of alluvial origin, whose periods range between the tertiary and the Quaternary periods. The rock formations include: Dammam, Euphrates, Fatha, Injana, Zahra, and Dibdibba Formation. Two of the sampling sites are located within the Dammam Formation, which consists of limestone, chalk and organic rocks, while the rest of the sites are located within the Dibdibba Formation, which is composed of brittle materials, clays and sandstones [17].

Page 2 line 93 Capitalize the first letter of a word at the beginning of a sentence.

Done. Thank you

Page 3 line 106 Add units for latitude and longitude data in Table 1.

Done. Thank you

Page 3 line 107 Figure 1 should be redrawn to better display the study area location and sampling sites information.

Done.

Page 4 line 127 ‘Groundwater is the primary source of water for activities of agricultural irrigation’. You need to specify the location, or it won't work.

Thank you. It deleted based on the opinion of another reviewer.

Page 4 line 118 Normalize the subscripts of symbols (Qi,Wi...).

Done. Thank you

Page 5 line 175 Add units for physical and chemical indicators in Table 4.

Done. Thank you

Page 6 line 185 The article indicates ‘All samples were distributed around the freshwater evaporation line in this figure’, but this conclusion is not visible in Figure 4.

Thank you for remark. there is a mistake in the figure.

Page 8 line 226 Use the same font size in your sentence.

Done. Thank you

Page 15 line 372 Keep your sentence writing standard.

Done. Thank you

There are some typos in the text; I warmly suggest to revise the English language.

Answer: We do agree with the reviewer comment therefore, we edit the English language by native English person.

Reviewer 2 Report

The research topic is relevant for the readership of Environment. However, the paper needs more elaboration before it can be finally published. It lacks a clear strategy for motivation of the work and the methodology is not entirely clear. Some figures need additional revision.

The presentation of the work needs improvement. The background information provided in the introduction section is too general. It is a general discussion of groundwater quality assessment, with references to many papers, but it is not focused on the specific topics analysed in the paper or the methods that are applied. In its current drafting, it does not help the reader to understand the motivation of the work and identify where the authors go beyond the state of the art with their work. I suggest rewriting this section, stating the main research questions addressed in the work and referencing the authors that have tackled these questions before, summarizing their results and identifying the gaps to be filled by this work.

I found the presentation of methodology a bit confusing. The methodology section presents the methods used, but sections 2.3.1 and 2.3.2 have the same subtitle. The authors should clearly state their objectives at the beginning and devote the methodological section to present their strategy, indicating how they are going to tackle the research questions addressed in their work and why they chose the methods they are proposing.

In addition, the manuscript is not properly finished. There are too many grammar and typographical errors that indicate that the manuscript is still at an early stage of revision. For instance, there are space (line 21) and extra punctuation (line 22) in the middle of the sentence. The authors should have checked their manuscript for such evident errors before submission. You may find below several of such errors, but the list is far from complete.

The manuscript should be polished.

Specific comments

Page 1 line 26 to 28 There is ambiguity in the expression of the sentence.

Page 1 line 32 Keywords are words or terms selected from reports and papers for literature indexing to express the full text topic content information, rather than the selection of individual target indicators. 

Page 2 line 63 The meaning of the text ‘The high-quality water covered nearly 90% of the region’ is not clear. Do you mean the study area?

It is better to add hydrogeological conditions of the study area in Section 2.1 Site Description.

Page 2 line 93 Capitalize the first letter of a word at the beginning of a sentence.

Page 3 line 106 Add units for latitude and longitude data in Table 1.

Page 3 line 107 Figure 1 should be redrawn to better display the study area location and sampling sites information.

Page 4 line 127 ‘Groundwater is the primary source of water for activities of agricultural irrigation’. You need to specify the location, or it won't work.

Page 4 line 118 Normalize the subscripts of symbols (Qi,Wi...).

Page 5 line 175 Add units for physical and chemical indicators in Table 4.

Page 6 line 185 The article indicates ‘All samples were distributed around the freshwater evaporation line in this figure’, but this conclusion is not visible in Figure 4.

Page 8 line 226 Use the same font size in your sentence.

Page 15 line 372 Keep your sentence writing standard.

There are some typos in the text; I warmly suggest to revise the English language.

Author Response

(The authors gave the same response as above.)

Round 2

Reviewer 1 Report

My comments and questions are well answered. Then I suggest to accept it at present version. 

Author Response

Cover Letter

Dear Editor,

I would like to submit a revised form of the manuscript entitled "Groundwater Hydrogeochemical and Quality Appraisal for Agriculture Irrigation in Greenbelt Area, Iraq" for consideration for publication in the Environments. This article is revised according to the reviewers’ comments and highlighted in different color in the revised manuscript. Please answer to the reviewers’ comments as in the attached report.

I appreciate the effort of the Reviewers to improve our article, thank you.

Your consideration for this manuscript with revised form is highly appreciated.

Sincerely

Prof. Dr. Qusay F. Alsalhy

Membrane Technology Research Unit

Chemical Engineering Department

University of Technology,

Alsinaa Street No. 52

Baghdad, Iraq

      [email protected]

Reviewer 2 Report

This manuscript is poorly recognized and  presented, I suggest "reconsider after major revision". 

1. Hydrogeology of the study area should be introduced in more detail.

2. Evaporation is not fully discussed. It is suggested that changes of TDS/EC or saturation index (SI) along groundwater flow path be examined.

3. Style of the figures shouled be improved; Fig.1, higher quality picture should be provided.

4. Line 17, “groundwater hydrogeochemical characteristics” should be “groundwater hydrogeochemical characteristics and processes”.

Author Response

Cover Letter

Dear Editor,

I would like to submit a revised form of the manuscript entitled "Groundwater Hydrogeochemical and Quality Appraisal for Agriculture Irrigation in Greenbelt Area, Iraq" for consideration for publication in the Environments. This article is revised according to the reviewers’ comments and highlighted in different color in the revised manuscript. Please answer to the reviewers’ comments as in the attached report.

I appreciate the effort of the Reviewers to improve our article, thank you.

Your consideration for this manuscript with revised form is highly appreciated.

Sincerely

Prof. Dr. Qusay F. Alsalhy

Membrane Technology Research Unit

Chemical Engineering Department

University of Technology,

Alsinaa Street No. 52

Baghdad, Iraq

      [email protected]

Comments and Suggestions for Authors

Reviewer 2

This manuscript is poorly recognized and presented, I suggest "reconsider after major revision". 

  1. Hydrogeology of the study area should be introduced in more detail.

Answer: We added new section (2.2. Geological and hydrogeological setting) and wrote more details as below:

The study area is part of the western plateau (the Najaf desert), bordered on the east by the Euphrates River and the alluvial plain, on the south by Lake Najaf, and on the north by the Karbala and Babil governorates. The study area consists of successive rock formations of alluvial origin, whose periods range between the tertiary and the Quater-nary periods. The rock formations of the study area include: Dammam, Euphrates, Fatha, Injana , Zahra, and Dibdibba Formations, as follows [17]:

  • Dammam Formation: It consists of limestone, chalk, and organic rocks. The Dammam Formation deposited in a coastal and continental environment with warm, highly saline waters. (2) Formation Euphrates: This formation consists of chalky limestone and sandy limestones, its thickness ranges from 10-16 m and the depositional environment are a shallow and warm environment. (3) Fatha Formation: it consists of sandy and calcareous rocks, its thickness between 10-15 m. The depositional environment of this formation is a coastal marine environment. (4) Injana Formation: it consists of a succession of clay rocks and layers of sandy rocks rich in calcareous carbonates, and its thickness about 35 m. (5) Zahra Formation: It consists of a succession of lime-stone and clay rocks or sandy and clay rocks. Its thickness reaches about 30 m. (6) Dibdibba Formation: It consists of fragile sediments that include a mixture of sand and gravel derived from igneous rocks, the thickness of the formation ranges from 2-10 m [17].

Two of the sampling sites are located within the Dammam Formation, while the other sites are located within the Dibdibba Formation.

The study area is considered one of the important areas from a hydrogeological view because it contains groundwater reservoirs represented by the sandy Dibdibba formations and limestone Dammam formation. The exposure of part of the sandy Dibdibba formation helped to renew its water and maintain groundwater storage in it through the penetration of rainwater and surface flood into it. The groundwater moves within the Dibdibba reservoir from the west towards the east and southeast in the region, and the quality of the groundwater in this reservoir is characterized by a high concentration of dissolved salts in it. The Dammam Formation is considered a water reservoir in most of the area, and the western hydraulic boundaries of this reservoir are areas of continuous groundwater movement coming from the west and southwest of the Western Desert. The groundwater in the Dammam limestone reservoir moves from the west towards Euphrates River. Its water is a mixture of old and newer water that comes from rain water in the previous rainy periods, and the salinity of the groundwater in this reservoir is of variable concentration [18].

  1. Evaporation is not fully discussed. It is suggested that changes of TDS/EC or saturation index (SI) along groundwater flow path be examined.

Answer:

As round 1, Depending on one reviewer's suggestion that "the subsection “3.1.2 evaporation” isn't directly related to the main topic of this study", and "the effect of evaporation on groundwater quality could be much low in study area because of the ions' distribution in groundwater is mainly influenced by evaporation when water table is high, which is generally 2-3 m below ground surface, and in study area was highest than this range", therefore we decided to delete this subsection.

Also, we explained the parameters TDS and EC in the salinity subsection, and we didn't explain the saturation index in our article because we didn't have full information to calculate this index.

  1. Style of the figures should be improved; Fig.1, higher quality picture should be provided.

Answer:

We changed (Figure 1) to high quality picture. Also, we changed the style for Figure 2 & 3, and Figure 6. We didn’t change the style of another figures because it gives the exact information that what we want in our article.4. Line 17, “groundwater hydrogeochemical characteristics” should be “groundwater hydrogeochemical characteristics and processes”.

Answer:

We changed line 17 to (groundwater hydrogeochemical characteristics and processes).
